# Comparative Investigation of Amino Acids Content in the Dry Extracts of *Juno bucharica, Gladiolus Hybrid Zefir, Iris Hungarica, Iris Variegata* and *Crocus Sativus* Raw Materials of Ukrainian Flora

**Olha Mykhailenko** [1] , **Liudas Ivanauskas** [2] , **Ivan Bezruk** [3] , **Roman Lesyk** [4,5,*] and **Victoriya Georgiyants** [3]

1   Department of Botany, National University of Pharmacy, 4 Valentinivska, 61168 Kharkiv, Ukraine; Mykhailenko.farm@gmail.com
2   Department of Analytical and Toxicological Chemistry, Lithuanian University of Health Sciences, A. Mickevičiaus 9, LT 44307 Kaunas, Lithuania; liudasivn@gmail.com
3   Department of Pharmaceutical Chemistry, National University of Pharmacy, 4 Valentynivska, 61168 Kharkiv, Ukraine; vania.bezruk@gmail.com (I.B.); vgeor@nuph.edu.ua (V.G.)
4   Department of Pharmaceutical, Organic and Bioorganic Chemistry, Danylo Halytsky Lviv National Medical University, 69 Pekarska, 79010 Lviv, Ukraine
5   Department of Public Health, Dietetics and Lifestyle Disorders, Faculty of Medicine, University of Information Technology and Management in Rzeszow, Sucharskiego 2, 35-225 Rzeszow, Poland
*   Correspondence: dr_r_lesyk@org.lviv.net; Tel.: +38-067-703-8010

**Abstract:** The aim of this research was the comparative study of the amino acids content in the dry extracts of *Iridaceae* plants of Ukrainian flora: *Juno bucharica* leaves and corms, *Gladiolus hybrid Zefir* leaves, *Iris hungarica* and *Iris variegata* rhizomes, and *Crocus sativus* stigmas, flowers, leaves and corms. A gas chromatography–mass spectrometry (GC–MS) method has been used. Separation of amino acids in the samples was carried out using a Shimadzu GC-MS-QP2010 equipped with an Rxi-5ms (Restek Corporation capillary column (30 m long, 0.25 mm outer diameter and 0.25 μm) with a liquid stationary phase (5% diphenyl and 95% polysiloxane) after derivatization with N-(t-butyldimethylsilyl)-N109 methyltrifluoroacetamide (MTBSTFA) reagent. The results obtained have shown that extracts from the aerial parts of plants investigated have a higher amino acid content and more diverse composition than the underground organs. Experimental data showed that *Crocus* leaves and *Juno* leaves extracts contain the highest general content of amino acids—31.99 mg/g and 14.65 mg/g respectively. All samples showed a high content of L-pyroglutamic acid (0.33–12.35 mg/g). Moreover, *Crocus* leaves and *Juno* leaves extracts had the most suitable amino acids composition and are prospective for further pharmacological studies.

**Keywords:** *Iridaceae*; dry extracts; amino acids; GC-MS; derivatization

## 1. Introduction

Nowadays, a number of questions about evidence-based phytotherapy are under discussion [1–4]. The main way to give scientific explanation to the pharmacological effects of natural pharmaceuticals is through the standardization of plant raw material and the recognition of the role of certain biologically active substances or their groups for pharmacological action. In accordance with modern concepts about the contribution of plant constituents to pharmacological activity, amino acids, as minor components,

are very important for different kinds of action [5–7]. Sometimes, the presence/absence of certain amino acids is the key factor influencing the central nervous system, thyroid gland, etc. [8,9].

Various plant species of the *Iridaceae* family are known for their magnificent decorative flowers as they are widely cultivated and wild-growing in Europe countries. At the same time, some *Iridaceae* plants are also known to be widely used as food, condiments or as medicinal plants [10–12]. Thus a high pharmacoligical potential was discovered for extracts and active constituents of plants from genes *Crocus, Iris, Gladiolus* and *Juno* [13–17]. Anti-inflammatory, phytoestrogenic, antidiabetic, hepatoprotective, hypolipidemic [10–12], antioxidant [13,14,17,18], antimicrobial [16,18] and anticancer [19,20] activities are described for some plants of these genes. *C. sativus* stigma [21], petal, stamen, and flower [22] extracts show good antioxidant activity in vitro antioxidant models due to the presence of carotenoids crocins and phenolic compounds (derivatives of kempferol and quercetin). Extracts of *C. sativus* leaves showed higher antioxidant activity than petals [15]. *Iris* species are characterized by a high content of xanthon mangiferin and different flavonoids and isoflavones (apigenin, kampferol, quercetin, vitexin, tectoridin, nigricin and its glucosides), which determine the presence of antioxidant [13,14,23] and antiviral activity [24]. Isoflavones and their glycosides isolated from rhizomes of *I. kashmiriana* [25] and *I. germanica* [19], as well as the extracts of *I. pseudopumila* and *I. florentina*, have been reported [10] to have cytotoxic properties. Phytochemical investigations showed that in these plants there is the presence of isoflavones [10,11,19], flavones [10,11,13,16,22], xanthones [10,14], quinones, peltogynoids and stilbenes [10,13], fatty acids [15], carotenoids [11,12,20], triterpenes, iridals [10,26]. However, the composition of the amino acids of these plants materials belongs to this family has not been studied much [27,28].

Separation and quantification of amino acids in a complex matrix such as herb material is a difficult task due to their high polarity with low volatilization and no chromophore groups. Furthermore, different techniques were developed and validated for their identification and quantification. Indirect methods are more popular and widely used. They include pre- or post-column derivatization with various reagents, for instance ninhydrin, ortho-phthalaldehyd (OPA) [29,30], phenyl isothiocyanate (PITC) [31], 2,4-dinitrofluorobenzene (DNFB), dimethylaminoazobenzene sulfonyl chloride (Dabsyl-Cl), 9-fluorenylmethyloxycarbonyl chloride (FMOC-Cl) [32], 9-fluorenylmethoxycarbonyl glycine chloride, and 6-aminoquinolyl-N-hydroxysuccinimidyl carbamate [33]. On other hand, direct methods are easier to apply and require no reagent. For instance, capillary electrophoresis equipped with an ultraviolet detector [34]—as well as a recently a new method of amino acids analysis—has been performed using UPLC-HILIC MS/MS method [35]. Furthermore, GC/MS is a promising method for the analysis of a wide range of amino acids due to remarkable chromatographic resolution combined with reproducible MS detection [36]. Some research [37,38] provides data on how the precision of quantification in analysis utilizing LC–MS can be compromised due to changes of ionization efficiency caused by matrix effects and the co-elution of metabolites. Such data hold true despite the presence of isotope-labeled internal standards, indicating a significant limitation to the development of quantitative untargeted LC/MS methods. LC methods are accurate enough and reliable but not eco-friendly. Furthermore, GC/MS provides the high sensitivity of the single quadrupole mass detector, reproducible ionization and fragmentation pattern of EI for a broad range of metabolite derivatives, informative mass spectra [39], as well as low maintenance cost [40,41].

In Ukrainian flora, the *Iridaceae* family is represented by many plants both harvested and wild-growing. Among them *Crocus sativus*, *Juno bucharica*, *Gladiolus hybrid Zefir*, *Iris hungarica* and *Iris variegata* are very interesting as prospective phytopharmaceuticals sources. Taking into account the possible role of amino acids as pharmacological activity, we decided to perform a comparative investigation into the content of these constituents in chosen plant raw material of *Iridaceae* family representatives from Ukrainian flora.

## 2. Materials and Methods

### 2.1. Materials and Methods

Analytical and chromatographic grade reagents were used and prepared according to the requirements of the State Pharmacopoeia of Ukraine [42] and European Pharmacopoeia [43]. Acetonitrile was purchased from Sigma-Aldrich GmbH (Karlsruhe, Germany). Purified water was produced by Millipore (Bedford, MA, USA) water purification system. A mix of L-amino standard acids (alanine, serine, valine, threonine, leucine, isoleucine, proline, aspartic acid, glutamic acid, lysine, methionine, phenylalanine, tyrosine) were purchased from Sigma-Aldrich GmbH (Steinheim, Germany). MTBSTFA was purchased from Sigma-Aldrich (St. Louis, MO, USA).

### 2.2. Plant Raw Materials

Leaves and corms of *Juno bucharica* (Foster) Vved (a voucher specimen CWU0056539), *Gladiolus hybrid Zefir* (a voucher specimen CWU0056538), *Iris hungarica* Waldst. et Kit. rhizomes (a voucher specimen CWU0056534), *Iris variegata* L. rhizomes (a voucher specimen CWU0056545) were harvested from the collection of botanical gardens of V. N. Karazin Kharkiv National University (Kharkiv, Ukraine) in May 2018. *Crocus sativus* (a voucher specimen CWU0056541-CWU005654) stigmas, flowers, leaves and corms were collected from a plantation in the village of Lyubimivka (Kahovka, Ukraine) in November 2017. All voucher specimens were verified by Yu. G. Gamulya and deposited at the Herbarium of V. N. Karazin Kharkiv National University, Kharkiv, Ukraine. The fresh plant material was air dried and crushed to obtain powder.

### 2.3. Preparation of Extracts

Powdered plant material (100 g) of *Juno* corm and leaves, *Gladiolus* and *Crocus* leaves, *Crocus* flowers and corms, and *Iris* species rhizomes were extracted with distilled water (1 L), on a water bath at 80 °C for 2 h three times, extracts were filtered and concentrated using a rotavapor and then completely dried in a drying cabinet and preserved at 4 °C until further use.

*C. sativus* stigma powder (5 g) was treated with hot distilled water (500 mL, 80 °C) and kept in a dark place for 24 h. After collecting the resulting extracts, the raw material was treated again with distilled water (500 mL, 4 °C) and kept in a dark place for 24 h. This maceration process was repeated a third time using 500 mL of distilled water at 4 °C and keeping the solution in a dark place for 24 h.

### 2.4. Sample Preparation and Derivatization

*Test solutions* were prepared as follows: 0.1 g of dry extracts were weighed into a volumetric flask and extracted with methanol (10 mL) in an ultrasonic bath at room temperature (20 ± 2 °C) for 15 min. The obtained solution was centrifuged for 10 min at 5000 rpm at 25 °C. An amount of 500 μL of supernatant was evaporated under nitrogen gas to dry residue. The resulting precipitate was dissolved in 100 μL of acetonitrile and 100 μL of MTBSTFA. This solution was heated at 100 °C for 2.5 h in a glycerol bath. Then, 1 μL of the test solution was injected into a gas chromatograph.

*Preparation of standard solutions.* An amount of 100 μL of the L-amino acids standards mixture (reference samples of alanine, serine, valine, threonine, leucine, isoleucine, proline, aspartic acid, glutamic acid, lysine, methionine, phenylalanine, tyrosine) were taken and dried under nitrogen gas to dry residue, then 100 μL of acetonitrile and 100 μL of MTBSTFA were added. The obtained solution was heated at 100 °C for 2.5 h in a glycerol bath.

### 2.5. GC–MS Analysis

The analyses were carried out using a SHIMADZU GC-MS-QP2010. The separation of amino acids was carried out using a Rxi-5ms capillary column (30 m long, 0.25 mm outer diameter and 0.25 μm liquid-stationary phase thickness) (Restek Corporation, Bellefonte, PA) with a liquid stationary

phase (5% diphenyl and 95% polysiloxane), and carrier gas helium. Oven temperature was set at 75 °C and kept for 5 min, and then increased at 10 °C/min to 290 °C, then at 20 °C/min to 320 °C and finally held at that temperature for 5 min. The injector temperature was 260 °C. The injection was made in the mode split (splitter 1:20) at 260 °C with an injection volume 1 μL. Mass spectra (70 eV, electron impact mode) scan range of m/z 35-500 amu with mass scan time of 0.2 s, interface temperature 280 °C. Flow column rate was 1.5 mL/min, pressure 100 kPa, total flow rate: 34.4 mL/min. The analytical run was in total 41 min. The content of identified amino acids is presented in Table 1.

### 2.6. Statistical Analyses

Statistical analysis was performed via a one-way ANOVA analysis of variance followed by Tukey's multiple comparison test by using the software package Prism v.5.04 (GraphPad Software Inc., La Jolla, CA, USA). The value of $p < 0.05$ was taken as the level of significance.

**Table 1.** The content of free amino acids in *Iridaceae* dry aqueous plant extracts, µg/g.

| N | R.Time | Name of Amino Acid | *Gladiolus* Leaves | *Juno* Leaves | *Crocus* Leaves | *Crocus* Flowers | *Crocus* Stigma | *Crocus* Corms | *Juno* Corms | *Iris Hungarica* Rhizome | *Iris Variegata* Rhizome |
|---|---|---|---|---|---|---|---|---|---|---|---|
| 1 | 14,98 | ʟ-Alanine | 34.9 ± 0.7 | 1032.2 ± 15.1 | 2709.5 ± 75.1 | 156.3 ± 1.7 | 60.0 ± 2.1 | 204.1 ± 2.1 | 52.1 ± 1.5 | n/d | 114.9 ± 1.4 |
| 2 | 16,51 | ʟ-Valine | 24.3 ± 0.5 | 1411.1 ± 20.7 | 3023.0 ± 51.1 | 105.8 ± 2.1 | n/d | 458.7 ± 19.3 | 24.5 ± 0.5 | n/d | 81.2 ± 2.6 |
| 3 | 17,00 | ʟ-Leucine | n/d | 1016.2 ± 18.5 | 2401.5 ± 75.2 | 72.2 ± 1.6 | n/d | 80.7 ± 1.3 | n/d | n/d | n/d |
| 4 | 17,34 | Isoleucine | n/d | 1049.2 ± 25.4 | 2322.5 ± 67.8 | 101.7 ± 4.8 | n/d | 174.9 ± 4.8 | 31.2 ± 0.6 | n/d | 50.2 ± 1.2 |
| 5 | 17,70 | 4-Aminobutanoic acid (recalculated to ʟ-Glutamic acid) | n/d | 255.6± 7.2 | 728.8 ± 29.6 | n/d | n/d | n/d | n/d | n/d | n/d |
| 6 | 17,81 | ʟ-Proline | n/d | 73.8 ± 1.9 | 4683.5 ± 171.0 | n/d | n/d | 2251.6 ± 82.5 | 49.6 ± 1.8 | n/d | 68.8 ± 2.4 |
| 7 | 19,84 | ʟ-Pyroglutamic acid (recalculated to ʟ-Glutamic acid) | 1695.2 ± 46.1 | 8538.8 ± 196.4 | 12347.5 ± 517.3 | 1158.4 ± 29.7 | n/d | 6318.5 ± 295.1 | 330.9 ± 6.4 | 1340.1 ± 56.7 | 783.9 ± 19.3 |
| 8 | 19,80 | ʟ-Methionine | n/d | n/d | n/d | n/d | 84.1 ± 2.4 | n/d | n/d | n/d | n/d |
| 9 | 20,24 | ʟ-Serine | n/d | 348.6 ± 3.7 | 790.6 ± 25.3 | 77.5 ± 1.7 | 14.2 ± 0.2 | 213.6 ± 8.6 | n/d | n/d | 24.2 ± 0.8 |
| 10 | 20,57 | ʟ-Threonine | n/d | 473.8 ± 9.6 | 1645.5 ± 27.5 | 63.9 ± 1.3 | n/d | 190.1 ± 5.4 | n/d | n/d | n/d |
| 11 | 21,25 | ʟ-Phenylalanine | n/d | 391.0 ± 9.1 | n/d | n/d | n/d | 68.4 ± 2.3 | n/d | n/d | n/d |
| 12 | 21,86 | ʟ-Aspartic acid | n/d | 63.1 ± 1.5 | 355.9 ± 9.7 | n/d | n/d | n/d | n/d | n/d | 31.4 ± 0.7 |
| 13 | 22,95 | ʟ-Glutamic acid | n/d | n/d | 381.8 ± 9.9 | n/d | n/d | n/d | n/d | n/d | n/d |
| 14 | 23,91 | ʟ-Lysine | n/d | n/d | 150.5 ± 40.9 | n/d | n/d | n/d | n/d | n/d | n/d |
| 15 | 26,08 | ʟ-Tyrosine | n/d | n/d | 452.7 ± 16.2 | n/d | 326.6 ± 6.9 | n/d | n/d | n/d | n/d |
| | | Total amount | 1757.1 | 14653.2 | 31993.3 | 1735.8 | 484.9 | 9960.5 | 488.3 | 1340.1 | 1154.6 |

n/d: not detected.

## 3. Results

### 3.1. Amino Acids Analysis

The GC–MS chromatograms of the test solutions established the presence of 13 amino acids and 2 additional amino acids (4-aminobutanoic and pyroglutamic acids), which were re-calculated on L-glutamic acid. Amino acids in *Iridaceae* plants dry extracts were identified by comparing the retention times of selected amino acids in specific MS chromatograms. Quantitative analysis was performed using calibration curves build with standards solutions. Concentration range of standards was from 0.1 to 100 ng/mL.

The elution order of amino acids standards is presented in Figure 1: alanine, valine, leucine, isoleucine, proline, metionine, serine, threonine, phenylalanine, aspartic acid, glutamic acid, lysine, tyrosine. The standards have been extracted and derivatized as previously described in the sample preparation session. Precision gave a lower value than RSD = 2%, except Arg, Cys and Tyr and sensitivity value lower than 10 ng of all amino acids injected. All standards samples followed the same extraction and derivatization steps. Additionally, two amino acids, viz. 4-aminobutanoic and L-pyroglutamic acids, were calculated via L-glutamic acid. The quantitative content of identified amino acids is presented in Table 1.

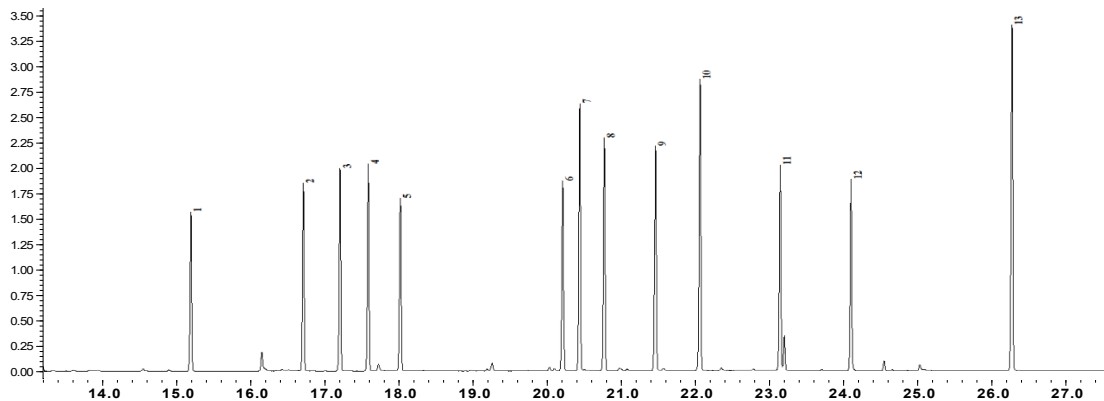

**Figure 1.** GC-MS chromatogram of amino acid standards mixture: 1) L-Alanine; 2) L-Valine; 3) L-Leucine; 4) Isoleucine; 5) L-Proline; 6) L-Methionine; 7) L-Serine; 8) L-Threonine; 9) L-Phenylalanine; 10) L-Aspartic acid; 11) L-Glutamic acid; 12) L-Lysine; 13) L-Tyrosine.

### 3.2. Method Validation

To validate the results obtained in our study, precision, repeatability, limit of detection (LOD), limit of quantification (LOQ), and linearity were calculated according to the International Conference for Harmonization (hereafter, ICH) [44]. The data obtained are presented in Tables 2–4. The method was validated using amino acid standards. The limit of detection (LOD) (signal/noise = 3) and the limit of quantification (LOQ) (signal/noise = 10) of all compounds varied within the range 0.14–0.72 ng/mL and 0.2–2.4 ng/mL respectively. The precision of the method was evaluated by measuring the peak chromatographic area of amino acids six times on the same test sample. The precision of retention times and peak areas of mixing amino acids standards for the replicated injection were in the range of 0.70–1.92% of RSD (n = 6). The reproducibility and repeatability of the method were evaluated by analysing six injections of the sample solution and two replicates of herbal sample extraction, respectively. To confirm the accuracy of the method, a recovery experiment was performed by mixed quantified samples with specific quantities of reference compounds. The average percentages of recovery of mixed amino acids standards ranged from 81.92% to 113.00%. The results demonstrated that conditions for the amino acids analysis were repeatable and accurate.

**Table 2.** Regression parameter, linearity, limit of detection (LOD) and limit of quantification (LOQ) of the proposed GC–MS method.

| Amino Acid | Calibration Curve | Correlation coefficient $r^2$ | Linear Range (μm/mL) | LOD (ng/mL) | LOQ (ng/mL) |
|---|---|---|---|---|---|
| L-Alanine | $f(x) = 276580.863935 \times x - 353207.105647$ | 0.983122 | 11.15–44.6 | 0.72 | 2.4 |
| L-Valine | $f(x) = 243032.088633 \times x + 49012.859468$ | 0.975345 | 14.7–58.8 | 0.585 | 1.930 |
| L-Leucine | $f(x) = 277306.790190 \times x - 217513.405090$ | 0.963603 | 16.5–58.8 | 0.1257 | 0.459 |
| Isoleucine | $f(x) = 228260.308590 \times x + 246041.176091$ | 0.969531 | 16.45–65.8 | 0.125 | 0.491 |
| L-Proline | $f(x) = 280385.380976 \times x + 1325395.667639$ | 0.972168 | 14.4–57.6 | 0.247 | 0.970 |
| L-Methionine | $f(x) = 187897.462370 \times x - 557659.056929$ | 0.981929 | 18.65–74.6 | 0.184 | 0.213 |
| L-Serine | $f(x) = 317111.017808 \times x + 1640589.305389$ | 0.957872 | 13.15–52.6 | 0.14 | 0.470 |
| L-Threonine | $f(x) = 249467.821934 \times x + 2284199.418438$ | 0.953095 | 14.9–59.6 | 0.132 | 0.44 |
| L-Phenylalanine | $f(x) = 206843.387916 \times x + 74215.145383$ | 0.963723 | 20.65–82.6 | 0.375 | 0.482 |
| L-Aspartic acid | $f(x) = 267570.147433 \times x + 2368216.421104$ | 0.942835 | 16.65–66.6 | 0.308 | 0.362 |
| L-Glutamic acid | $f(x) = 214747.719667 \times x - 596641.230769$ | 0.980993 | 18.4–73.6 | 0.195 | 0.651 |
| L-Lysine | $f(x) = 232710.756329 \times x - 3270542.068621$ | 0.988024 | 18.25–73.00 | 0.34 | 0.144 |
| L-Tyrosine | $f(x) = 213345.354626 \times x + 3840401.269116$ | 0.885138 | 22.65–90.6 | 0.53 | 1.76 |

**Table 3.** Precision and repeatability of GC–MS method.

| Amino Acid | RSD of Retention Time (%) | Precision | Repeatability |
|---|---|---|---|
| | | RSD of Area (%) | RSD of Area (%) |
| L-Alanine | 0.01 | 1.52 | 0.75 |
| L-Valine | 0.009 | 1.34 | 1.64 |
| L-Leucine | 0.01 | 1.78 | 1.70 |
| Isoleucine | 0.008 | 1.80 | 0.83 |
| 4-Aminobutanoic acid | 0.01 | 1.92 | 1.73 |
| L-Proline | 0.01 | 0.78 | 0.95 |
| L-Pyroglutamic acid | 0.1 | 1.71 | 0.94 |
| L-Methionine | 0.008 | 1.53 | 1.72 |
| L-Serine | 0.008 | 1.94 | 0.62 |
| L-Threonine | 0.01 | 0.83 | 1.48 |
| L-Phenylalanine | 0.009 | 1.71 | 0.72 |
| L-Aspartic acid | 0.01 | 2.39 | 1.37 |
| L-Glutamic acid | 0.009 | 1.56 | 1.41 |
| L-Lysine | 0.01 | 0.70 | 0.67 |
| L-Tyrosine | 0.04 | 1.75 | 0.88 |

**Table 4.** Accuracy of GC–MS method.

| Amino Acid | Found (µg/mL) | Added (µg/mL) | Recovery (%) | Amino Acid | Found (µg/mL) | Added (µg/mL) | Recovery (%) |
|---|---|---|---|---|---|---|---|
| L-Alanine | 10.69 | 11.15 | 95.86 | L-Threonine | 14.88 | 14.90 | 99.87 |
| | 19.40 | 22.30 | 86.99 | | 30.51 | 37.25 | 81.91 |
| | 34.16 | 33.45 | 102.11 | | 50.51 | 44.70 | 113.00 |
| L-Valine | 13.98 | 14.70 | 95.11 | L-Phenylalanine | 18.16 | 20.65 | 87.92 |
| | 25.66 | 29.40 | 87.27 | | 35.63 | 41.30 | 86.27 |
| | 46.95 | 44.10 | 106.46 | | 68.05 | 61.95 | 109.85 |
| L-Leucine | 15.96 | 16.50 | 96.74 | L-Aspartic acid | 14.58 | 16.65 | 87.56 |
| | 27.15 | 33.10 | 82.01 | | 34.77 | 33.30 | 104.42 |
| | 48.91 | 44.10 | 110.91 | | 56.61 | 49.95 | 113.34 |
| Isoleucine | 16.05 | 16.45 | 97.57 | L-Glutamic acid | 15.66 | 18.40 | 85.11 |
| | 28.17 | 32.90 | 85.63 | | 31.65 | 36.80 | 86.02 |
| | 54.87 | 49.35 | 111.18 | | 57.23 | 55.20 | 103.67 |
| L-Proline | 13.79 | 14.40 | 95.73 | L-Lysine | 16.39 | 18.25 | 89.82 |
| | 28.20 | 28.80 | 97.92 | | 25.93 | 27.38 | 94.72 |
| | 48.28 | 43.20 | 111.75 | | 51.51 | 54.75 | 94.08 |
| L-Methionine | 16.58 | 18.65 | 88.89 | L-Tyrosine | 18.67 | 22.65 | 82.43 |
| | 31.86 | 37.30 | 85.41 | | 49.89 | 45.30 | 110.13 |
| | 57.05 | 55.95 | 101.97 | | 79.25 | 67.95 | 116.62 |
| L-Serine | 11.93 | 13.15 | 90.73 | | | | |
| | 26.30 | 26.30 | 100.00 | | | | |
| | 44.53 | 39.45 | 112.89 | | | | |

## 4. Discussion

Due to the polar nature of amino acids, derivatization is required prior to GC analysis [45]. A higher sensitivity can be reached during the analysis of amino acids for GC-MS by using derivatization agents, such as N-tert-butyldimethylsilyl- N-methyltrifluoroacetamide (MTBSTFA) and N,O-bis(trimethylsilyl)trifluoroacetamide (BSTFA). Although, MTBSTFA derivatives are more stable and contain less moisture.

Previous good results using the MTBSTFA reagent was the reason for its use in our investigation. The suitability of this method—both for the identification and the quantification of amino acids in samples—was demonstrated by the good resolution (Figure 1) and validation results data (Tables 2–4).

Experimental data showed that the *Crocus* and *Juno* leaves extracts are characterized by the highest general content of amino acids—31.99 mg/g and 14.65 mg/g, respectively, compared with other *Iridaceae* dry extracts. The results of free amino acid content in *Iridaceae* extracts are shown in Table 1.

According to a comparative analysis of the composition of amino acids in examined *Iridaceae* extract plants parts, it can be seen that the aboveground organs (leaves, flowers, stigma) have a higher content and a diverse composition of amino acids than their underground organs (corms and rhizomes). The largest number of amino acids with high abundance were detected in the *Crocus* leaves extract (Figure 2a) and in *Juno* leaves dry extract (Figure 2b). Common among them were alanine, valine, leucine, isoleucine, proline, serine, threonine, aspartic acid, 4-aminobutanoic acid, and pyroglutamic acid. Additionally, the extracts of *Crocus* leaves contained glutamic acid, lysine, tyrosine. A significantly less amount and content of amino acids were found in *Gladiolus* leaves, *Crocus* flowers and *Crocus* stigma extracts. *Crocus* corms were found to have the highest amino acid concentration compared to the other underground organ extracts. The similar determination of free amino acids in the methanol extracts of *Acacia* and *Eucalyptus* leaves by GC–MS method with TBDMS derivatives [46] demonstrated that only a few free amino acids were identified including proline, methionine, phenylalanine, cysteine, and lysine. By the other investigation [47], arginine, leucine and glutamic acid in tubers and leaves of *Coccinia abyssinica* had the highest content among all amino acids. Besides, the average amino acid content was higher in the leaves compared to the tubers, that is

corresponding to the current investigation of *Iridaceae* plant extracts. Thus, studies are shown that the number of identified free amino acids in plant extracts is not high.

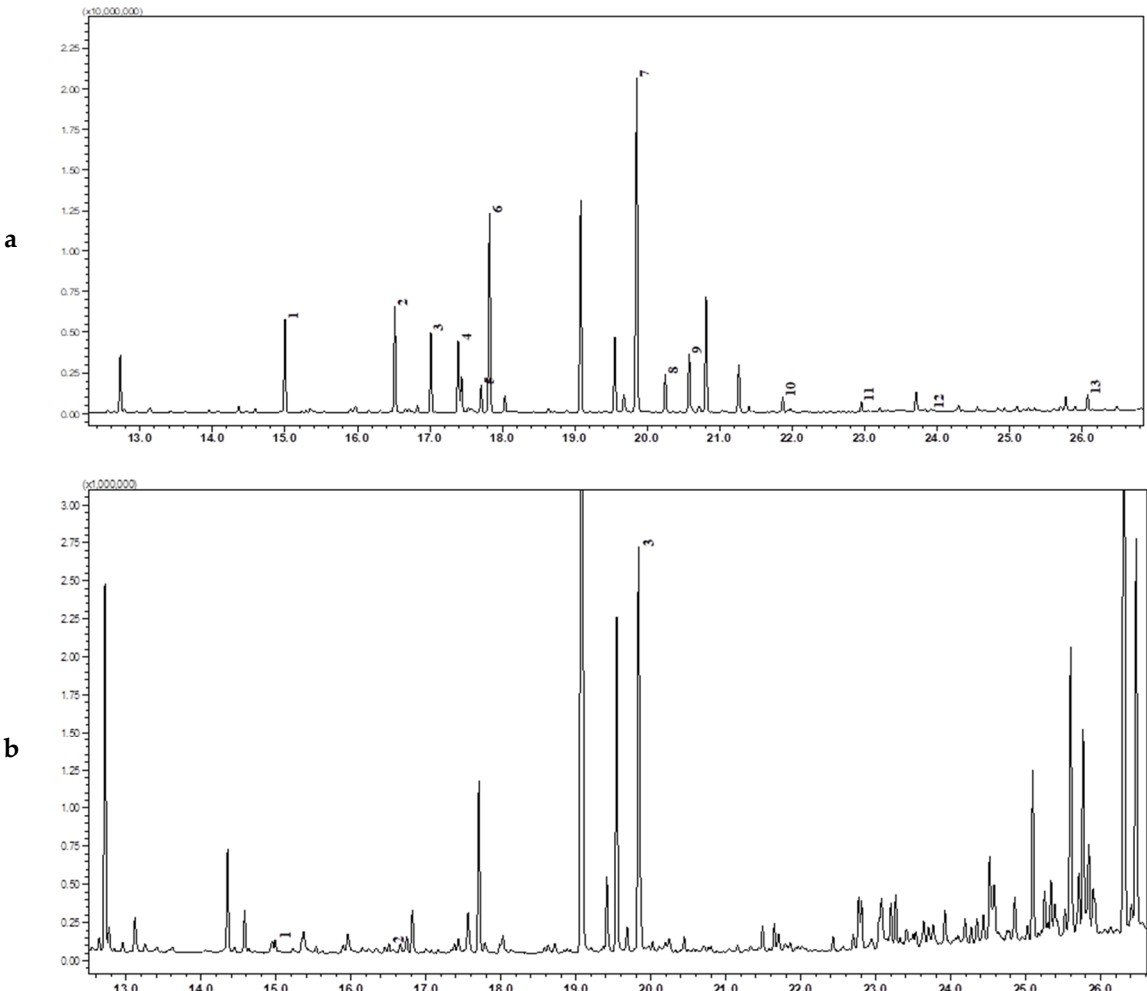

**Figure 2.** GC–MS chromatograms for the analysis of amino acid *Crocus* leaves (**a**) and *Gladiolus* leaves (**b**) dry extracts: 1) L-Alanine; 2) L-Valine; 3) L-Leucine; 4) Isoleucine; 5) L-Proline; 6) L-Methionine; 7) L-Serine; 8) L-Threonine; 9) L-Phenylalanine; 10) L-Aspartic acid; 11) L-Glutamic acid; 12) L-Lysine; 13) L-Tyrosine.

In *Iris hungarica* rhizomes, this method was able to identify only one pyroglutamic acid (1.34 mg/g), as well as in *Iris variegata* rhizomes extract alanine, valine, isoleucine, proline, serine, aspartic acid and also pyroglutamic acid were found. Corms extracts, viz. *Crocus* and *Juno* have five common amino acids; among them, pyroglutamic acid (6.32 mg/g and 0.33 mg, respectively) and proline (2.25 mg/g and 0.05 mg/g, respectively) were dominant.

The aboveground organs of *Iridaceae* were also characterized by a high content of pyroglutamic acid, from 1.16 mg/g for *Crocus* flower and up to 12.35 mg/g for *Crocus* leaves. The group of aliphatic amino acids such as alanine (0.03 to 2.71 mg/g), valine (0.02 to 3.02 mg/g), leucine (0.07 to 2.40 mg/g), and isoleucine (0.1 to 2.32 mg/g) were also found to be the highest compared to the other amino acids in samples.

The content of free amino acids for *C. sativus* extracts decreases at leaves > corms > flowers> stigmas. The results show that the dry aqueous extract of *Crocus* leaves has the highest concentration of certain free amino acids, which possibly explains [48] the presence of a pronounced antioxidant effect in leaves [17,21,22]. The predominant amino acids *Crocus* leaves extract were of the hydrophobic amino acids group (alanine (2.71 mg/g), valine (3.02 mg/g), leucine (2.40 mg/g), isoleucine (2.32 mg/g), proline

(4.68 mg/g)), which, according to the authors of [49], can inhibit free radicals by proton donation [50] and exhibited a high antioxidant activity in plant extracts. Among the investigated samples, the dry extract of *Crocus* leaves was characterized by the highest content of pyroglutamic acid (12.35 mg/g). The content of pyroglutamic acid in the *Crocus* raw material can be placed in the following order: leaves (12.35 mg/g) >corm (6.32 mg/g) > flower (1.16 mg/g). Besides this, the presence of glutamine acid (0.38 mg/g) and the essential amino acid lysine (0.15 mg/g) were found only in *Crocus* leaves extract. In the stigma extract only alanine, serine and tyrosine have been detected, as well as methionine (0.08 mg/g), which was found only in this sample. An analysis [51] of the free amino acids composition of 20 *Crocus* stigma from different countries (Spain, Italy, Greece, Iran) showed that alanine, proline and aspartic acid were the major amino acids in all tested samples. The differences in the obtained results can be explained by the environmental factors, that affect the plant chemical composition [52–55]. In *Crocus* stigma and leaves extracts tyrosine was also found (0.32 mg/g and 0.45 mg/g, respectively), which is absent in other *Iridaceae* extracts. Phenylalanine has been found only in *Crocus* corms and in small amounts (0.06 mg/g).

The *Juno* leaves extract has a more diverse amino acid composition compared to *Juno* corms. The content of all identified amino acids in *Juno* the corms extract is much less than in its leaves. 4-Aminobutanoic acid (0.25 mg/g) was identified in *Juno* leaves like in *Crocus* leaves (0.72 mg/g). Serine (0.34 mg/g), threonine (0.47 mg/g) and phenylalanine (0.39 mg/g) has been found only in *Juno* leaves as in *Crocus* corms. Among the hydrophobic amino acids in *Juno* leaves extract were alanine (1.03 mg/g), valine (1.41 mg/g), leucine (1.01 mg/g), isoleucine1.04 mg/g), proline (0.07 mg/g), and phenylalanine (0.39 mg/g).

In *Gladiolus* leaves, alanine (0.03 mg/g), valine (0.02 mg/g) and pyroglutamic acid (1.69 mg/g) were found in lower concentrations compared other extracts.

The major free amino acids in *Iridaceae* extracts were pyroglutamic acid (with the ranges of 0.3–12.35 mg/g), proline (with the ranges of 0.05–4.68 mg/g) and valine (with the ranges of 0.02–3.02 mg/g). Pyroglutamic acid has been described as the major amino acid in all extracts. According to authors [27,28] $\gamma$-glutamyl peptides are very common in *Iridaceae* plants.

The high content of pyroglutamic and aspartic acids in samples can be explained by the relationship between the primary and secondary metabolism of amino acids in plants, since glutamic acid is a precursor in the biosynthesis of all amino acids in plant parts [56,57]. Aspartic acid is synthesized by direct amination, and alanine and glutamic acid are formed as a result of reductive amination. All other amino acids are secondary ones, because they are formed as a result of the transamination of the amino acids listed above with the corresponding keto acids that arise during the metabolism, as well as by the conversion of some acids to others [58]. The dehydration of glutamic acid leads to the formation of its lactam—pyroglutamic acid, which is the key precursor in the biosynthesis of biologically active compounds of primary and secondary metabolites [59].

It should be noted that pyroglutamic acid plays a great role as a mediator of the central nervous system [58]. Thus, in the future, it would be advisable to pay attention to the examination of *Crocus* and *Juno* leaves extracts as possible CNS-agents. In addition, it should be noted that the content of essential amino acids [60] is high enough for the extracts of *Crocus* and *Juno* leaves. The highest content of valine (1.41–3.02 mg/g), leucine (1.01–2.40 mg/g), isoleucine (1.04–2.32 mg/g) and threonine (0.47–1.64 mg/g) for extracts were established. Methionine (0.08 mg/g) was identified only in the *Crocus* stigma extract, while lysine (0.15 mg/g) only in *Crocus* leaves extract. The content of essential amino acids also could be taken into account in future pharmacological studies.

As a result of this work, the comparative analyses of amino acids in the aqueous dry extracts of *Crocus sativus* (stigma, flowers, leaves, corms), *Juno bucharica* (leaves, corms), *Gladiolus hybrid Zefir* (leaves) and *Iris hungarica* (rhizomes), *Iris variegata* (rhizomes) from Ukrainian flora using a simple and sensitive gas chromatography method were carried out for the first time. It should be noted that the amino acid content in extracts from the aboveground organs (leaves, flowers, stigma) is much higher compared to their underground organs (corms and rhizomes). Pyroglutamic acid—as

well as valine, alanine, proline, valine, leucine, and isoleucine—predominate in all studied extracts. Due to the fact that amino acids take part in various metabolic processes and their role in pathological correction, it is important to learn about the pharmacological activities of investigated *Iridaceae* plants from Ukrainian flora. Attention should be paid to future pharmacological studies on the nootropic and moderate excitatory activities of the plant extracts examined. Furthermore, the predominant hydrophobic amino acid group (alanine, valine, leucine, isoleucine, proline, and phenylalanine) in the extracts is an important factor, that affects the antioxidant properties of plants.

**Author Contributions:** Conceptualization was done by O.M. and V.G.; Methodology and experimental works were done by O.M., I.B. and L.I.; Data Analysis was done by L.I., O.M. and R.L.; writing, review and editing the paper were done by O.M., I.B. and V.G., Project administration and supervision were done by R.L. and V.G. All authors have read and agreed to the published version of the manuscript.

**Funding:** This research received no external funding.

**Conflicts of Interest:** The authors declare no conflict of interest.

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
