# Peer review of "Comparative Investigation of Amino Acids Content in the Dry Extracts of Juno bucharica, Gladiolus Hybrid Zefir, Iris Hungarica, Iris Variegata and Crocus Sativus Raw Materials of Ukrainian Flora"

_scipharm, doi:10.3390/scipharm88010008_

Round 1

Reviewer 1 Report

The article of Mykhailenko et al reports a "Comparative investigation of amino acids content in dry extracts of Juno bucharica, Gladiolus hybrid Zefir, Iris hungarica, Iris variegata and Crocus sativus raw materials of Ukrainian flora ".
I have almost nothing to comment on how the work has been motivated and performed.

Major comments:

My main comments are on the introduction and conclusion.
Similar studies have probably been reported on other natural plants. It would be very useful to recall these studies and the authors' conclusions. Likewise, if the comparisons of the conclusion between the different parts of the plants are interesting, a comparison with completely different plants would be useful. Are these results really specific to the plants studied or really usual? 

I cannot understand the absence of glycine. I thought it must be the most abundant. Please give a clear explanation about that.

Minor comments:

Line 92: "was extracted with distilled water (1 L), on a water bath for 2 hours for three times". Could you indicate the temperature?

Why is the experimental procedure for C. sativus stigma different?

Some typos

Author Response

Dear Reviewer 

We thank for positive feedback and we are grateful for careful reading of our manuscript and for the constructive and supportive comments which improved it. Repeated work was carried out with the manuscript; all clarifications and corrections were made.

My main comments are on the introduction and conclusion. Similar studies have probably been reported on other natural plants. It would be very useful to recall these studies and the authors' conclusions. Likewise, if the comparisons of the conclusion between the different parts of the plants are interesting, a comparison with completely different plants would be useful. Are these results really specific to the plants studied or really usual?

We added to the comparison text the results of a similar analysis for other plants according to published data in Dissection section (line 221-228; 251-257). For these plants, similar studies were conducted for the first time and the results are of scientific interest.

I cannot understand the absence of glycine. I thought it must be the most abundant. Please give a clear explanation about that.

Glycine was not determined in the extracts by preliminary analysis using paper chromatography. Therefore, it was decided not to include in the GC-MS analysis the amino acids such as glycine, cysteine, leucine, asparagine, glutamine, histidine, arginine, tryptophan, and cystine that were also not identified.

Yours faithfully

Roman Lesyk

Reviewer 2 Report

All my comments and suggestions are shown in the manuscript.

Author Response

We are very grateful to Reviewer 2 for a detailed analysis of our manuscript and an assistance in correcting the language of manuscript. We corrected all your comments, added the necessary sources of literature and discussion into “Discussion”, removed “conclusion” so that the material would not be repeated. All corrections and additions are marked in the file "scipharm-682232-original_correct”. And of course, a clean manuscript is framed - the file is " scipharm-682232-original_clear."

Round 2

Reviewer 1 Report

The article entitled "Comparative investigation of amino acids content in  dry extracts of Juno bucharica, Gladiolus hybrid Zefir, Iris hungarica, Iris variegata and Crocus sativus raw materials of Ukrainian flora" of Olha Mykhailenko et al. has been revised and the quality of the paper has improved significantly.

Several parts of the articles are clearer and many references have been added.

Only "Glycine was not determined in the extracts by preliminary analysis using paper chromatography" is still strange for me but I understand for the other missing amino acids.

To publish in present form. 

Author Response

Dear reviewer!

We would like to thank You for revision and constructive comments that helped significantly improve the manuscript.

Reviewer 2 Report

Dear Authors,

I tried to improve once again the english in your manuscript. It would be better if a native English speaker checks the entire work!

The version reviewed should be follewed.

Author Response

Dear Reviewer!

We would like to thank You for revision and constructive comments that helped significantly improve the manuscript. Your suggestions have been incorporated in the revised manuscript (green highlight).

We are very grateful for a detailed analysis of our manuscript and an assistance in correcting the language of manuscript. We conducted an additional analysis of the English language of the manuscript and made corrections.